# Development and Validation of the Artificial Intelligence in Mental Health Scale: Application for AI Mental Health Chatbots

**DOI:** 10.3390/healthcare13243269

**Published:** 2025-12-12

**Authors:** Aglaia Katsiroumpa, Olympia Konstantakopoulou, Ioannis Moisoglou, Parisis Gallos, Olga Galani, Paschalina Lialiou, Maria Tsiachri, Petros Galanis

**Affiliations:** 1Clinical Epidemiology Laboratory, Faculty of Nursing, National and Kapodistrian University of Athens, 10679 Athens, Greece; aglaiakat@nurs.uoa.gr (A.K.); olykonstant@nurs.uoa.gr (O.K.); mtsiachri@nurs.uoa.gr (M.T.); 2Faculty of Nursing, University of Thessaly, 38241 Larissa, Greece; iomoysoglou@uth.gr; 3Faculty of Nursing, University of West Attica, 12243 Egaleo, Greece; parisgallos@uniwa.gr (P.G.); ogalani@uniwa.gr (O.G.); 4Computational Biomedicine Research Lab, Department of Digital Systems, University of Piraeus, 18534 Piraeus, Greece; plialiou@unipi.gr

**Keywords:** artificial intelligence, chatbots, mental health, attitudes, validation, scale, tool, questionnaire

## Abstract

**Highlights:**

**What are the main findings?**
The Artificial Intelligence in Mental Health Scale (AIMHS) is a newly developed tool to measure people’s attitudes toward the use of Artificial Intelligence-based chatbots for mental health support.Validity and reliability analyses show that the AIMHS has robust measurement properties.

**What are the implications of the main findings?**
Comprising five items and requiring approximately two minutes to administer, the AIMHS represents a concise and user-friendly instrument for evaluating attitudes toward the use of Artificial Intelligence-based chatbots.Assessing individuals’ acceptance of AI-based mental health chatbots is critical for understanding their potential integration into mental healthcare services.

**Abstract:**

**Background/Objectives**: Artificial intelligence (AI)-based chatbots present a viable approach to overcoming several challenges associated with conventional psychotherapy, such as high financial costs, limited access to mental health professionals, and geographical or logistical barriers. Thus, these chatbots are increasingly employed as complementary tools to traditional therapeutic practices in mental health care. Our aim was to develop and validate a scale to measure attitudes toward the use of AI-based chatbots for mental health support, i.e., the Artificial Intelligence in Mental Health Scale (AIMHS). **Methods**: A multidisciplinary panel of experts assessed the content validity. To confirm face validity, we carried out cognitive interviews and calculated the item-level face validity index. We applied factor analysis to verify the construct structure. We assessed measurement invariance across demographic subgroups. Concurrent validity was evaluated using three valid instruments. Reliability was tested through Cronbach’s alpha, Cohen’s kappa, and the intraclass correlation coefficient. **Results**: Factor analysis supported a two-factor five-item model. The two factors were technical and personal advantages, and explained 81.28% of the variance. The AIMHS demonstrated adequate concurrent validity, evidenced by statistically significant correlations with Artificial Intelligence Attitude Scale (r = 0.405, *p*-value < 0.001), Attitudes Towards Artificial Intelligence Scale (acceptance subscale; r = 0.401, *p*-value < 0.001, fear subscale; r = −0.151, *p*-value = 0.002), and Short Trust in Automation Scale (r = 0.450, *p*-value < 0.001). Configural, metric and scalar invariance were supported by our findings. Cronbach’s alpha was 0.798, and intraclass correlation coefficient was 0.938. Cohen’s kappa for the five items ranged from 0.760 to 0.848. **Conclusions**: The AIMHS is a five-item psychometrically sound and user-friendly instrument capturing two dimensions; technical and personal advantages. Future research should be undertaken to further evaluate the psychometric properties of the AIMHS across diverse populations and contexts.

## 1. Introduction

Artificial Intelligence (AI), in its broadest sense, refers to the field of computer science focused on creating systems capable of performing tasks that typically require human intelligence [1]. As AI technologies continue to expand across diverse sectors, their integration into mental health services has become an increasingly important area of investigation [2,3].

Recent research has increasingly examined the use of AI chatbots as innovative tools for delivering digital mental health interventions. AI chatbots are increasingly being used in mental health support as complementary tools to traditional therapy [4]. These chatbots employ machine learning algorithms and natural language processing to interpret user input and generate contextually appropriate responses, thereby enabling interactions that more closely resemble human conversation [5]. AI chatbots offer a promising solution to several barriers commonly associated with traditional psychotherapy, including financial cost, limited availability of mental health professionals, and geographic or logistical constraints. Moreover, the anonymity and privacy afforded by chatbot platforms may reduce the stigma often linked to seeking mental health support [6,7].

Empirical studies examining the effectiveness of AI-based interventions have produced encouraging results. For instance, a systematic review of AI-assisted psychotherapy for adults experiencing symptoms of depression and anxiety revealed significant improvements in depressive outcomes [8]. Several studies have demonstrated reductions in both depression and anxiety symptoms among users of AI chatbots designed to provide mental health support [9,10,11,12,13,14,15]. AI chatbots may be particularly well-suited for young adults, a demographic that frequently uses online messaging platforms with interfaces similar to those of chatbot applications [7]. Moreover, young adults have demonstrated high levels of satisfaction with digital mental health resources, suggesting that chatbot-based interventions may be both accessible and appealing to this population [16].

Although preliminary findings support the effectiveness of AI chatbots and highlight their potential utility, empirical research on individuals’ attitudes toward these technologies remains limited. Attitude is commonly defined as a psychological predisposition expressed through evaluative judgments—ranging from favorable to unfavorable—toward a specific entity [17]. It may also be conceptualized as an affective orientation toward or against a psychological object. Attitudes serve adaptive functions and are instrumental in shaping behavior [18]. Given that technological innovations often elicit distinct attitudinal responses, it is imperative to understand public perceptions of AI-based applications, particularly in domains where such technologies are actively being implemented [19]. Consequently, the accurate assessment of attitudes through psychometrically sound instruments is essential for predicting behavioral responses and informing the future trajectory of technological developments, such as AI-driven mental health interventions.

To date, several scales have been developed to measure attitudes toward AI as a more overarching concept, such as the Attitudes Towards Artificial Intelligence Scale (ATTARI-12) [20], the Attitudes Towards Artificial Intelligence Scale (ATAI) [21], the General Attitudes Towards Artificial Intelligence Scale (GAAIS) [22], the Concerns About Autonomous Technology questionnaire [23], the Artificial Intelligence Attitude Scale (AIAS-4) [24], and the Artificial Intelligence Attitudes Inventory (AIAI) [25]. Additionally, several other scales have been developed to assess attitudes toward specific domains of AI such as the Attitudes Towards Artificial Intelligence at Work (AAAW) [26], the Treats of Artificial Intelligence Scale (TAI) [27], the Artificial Intelligence Anxiety Scale (AIAS) [28], the Short Trust in Automation Scale (S-TIAS) [29], and the Attitude Scale Towards the Use of Artificial Intelligence Technologies in Nursing (ASUAITIN) [30].

Evaluating individuals’ attitudes toward applications of AI in mental health has become increasingly relevant for both research and practice. Given the well-established relationship between attitudinal dispositions and help-seeking intentions [31,32,33], rigorous investigation of user perceptions toward chatbot-assisted psychotherapy is critical for evaluating its viability as an empirically supported mental health treatment modality. In this context, measuring attitudes toward the use of AI-based chatbots for mental health support is highly important, especially as these technologies become more integrated into clinical and therapeutic settings.

However, to the best of our knowledge, no existing scale measures individuals’ attitudes toward the use of AI-based chatbots for mental health support. The development of a psychometrically robust, concise measurement scale to assess attitudes toward AI-enabled chatbots in mental health applications would significantly advance empirical research and applied practices in evaluating and addressing public perceptions of artificial intelligence. Therefore, the aim of our study was to develop and validate a scale to measure attitudes toward the use of AI-based chatbots for mental health support, i.e., the Artificial Intelligence in Mental Health Scale.

## 2. Materials and Methods

### 2.1. Development of the Scale

AI chatbots for mental health support present distinct technical advantages that enhance their applicability in clinical and research contexts. These systems employ advanced natural language processing techniques to interpret user input and generate contextually appropriate responses, facilitating a conversational interface that mimics human interaction. They utilize machine learning algorithms to adapt responses based on user behavior and historical interactions, improving accuracy and relevance over time. Their continuous availability ensures uninterrupted support, overcoming temporal and geographical barriers inherent in traditional mental health services. Furthermore, through machine learning algorithms, chatbots dynamically adapt to individual user patterns, improving predictive accuracy over time. Integration with data analytics frameworks allows for real-time monitoring of behavioral indicators and early detection of psychological distress, thereby supporting preventive care strategies. Finally, their cloud-based infrastructure ensures scalability and compliance with data protection regulations, while maintaining secure storage and interoperability with electronic health record systems [19,34,35,36]. Thus, we considered that technical advantages could be a factor in our scale. In this context, we created items that refer to technical advantages of the AI chatbots such as “Artificial intelligence chatbots can demonstrate better problem-solving skills compared to a human therapist” and “Artificial intelligence chatbots can offer up to date information on mental health issues”.

Additionally, AI chatbots for mental health support offer significant personal benefits that enhance accessibility and user engagement. They provide immediate, on-demand support, reducing barriers such as scheduling constraints and geographic limitations commonly associated with traditional therapy. Their anonymity and privacy features foster a safe environment for individuals who may experience stigma or discomfort in seeking face-to-face care, thereby encouraging disclosure and emotional expression. Moreover, chatbots deliver personalized interventions by analyzing user input and tailoring responses to individual needs, which promotes a sense of relevance and trust. The non-judgmental nature of AI interactions further alleviates anxiety associated with human evaluation, making them particularly valuable for individuals with social apprehension [37,38,39,40,41]. Thus, we considered that personal advantages could be a factor in our scale. In this context, we created items that refer to personal advantages of the AI chatbots such as “Artificial intelligence chatbots can expand access to mental health care by reducing geographic barriers” and “Artificial intelligence chatbots can expand access to mental health care by reducing financial barriers”. On the other hand, AI chatbots can act as psychological stressors, generate hallucinations, and contribute to confusion or misinformation. Early evidence indicates that prolonged interaction with chatbots may exacerbate mental health vulnerabilities, leading to emotional dependence, increased loneliness and lower socialization [42]. Moreover, a recent thematic analysis of a dataset of 35,105 negative reviews indicates that users often encounter unwanted sexual advances, ongoing inappropriate conduct, and instances where the chatbot fails to honor personal boundaries. Many users reported feeling uncomfortable, experiencing privacy breaches, and feeling let down—especially when their goal was to engage with an AI companion in a platonic or therapeutic context [43]. AI chatbots often fail to detect subtle signs of distress and may inadvertently validate harmful thoughts, posing risks during mental health crises [44,45]. Thus, we considered that there are personal disadvantages concerning AI chatbots. In this context, we created items that refer to personal advantages of the AI chatbots such as “Artificial intelligence chatbots cannot provide correct information on mental health issues”, “Artificial intelligence chatbots cannot achieve empathy levels comparable to those of a human therapist”, and “Artificial intelligence chatbots cannot improve people’s mood”.

To further establish a solid framework for the development of our scale, we employed the Technology Acceptance Model which is the most widely utilized model to explore users’ attitudes towards new e-technology or new e-services [46,47]. In brief, the Technology Acceptance Model is based on the belief that perceived usefulness and perceived ease of use are the key determinants. Perceived usefulness is defined as the extent to which an individual believes that employing a specific system will enhance his/her performance. Within the context of this study, this construct reflects the perceived utility of AI chatbots for mental health support. Thus, we created items that refer to usefulness or, otherwise, utility of the AI chatbots such as “Artificial intelligence chatbots cannot appropriately respond to mental health emergencies (e.g., suicidal thoughts)” and “Artificial intelligence chatbots can demonstrate better problem-solving skills compared to a human therapist”. Additionally, perceived ease of use refers to the degree to which an individual believes that interacting with a system will require minimal effort. In this study, this concept pertains to the accessibility and user-friendliness of AI mental health chatbots. Thus, we created items that refer to ease of use or, otherwise, accessibility of the AI chatbots such as “Artificial intelligence chatbots can expand access to mental health care by reducing geographic barriers” and “Artificial intelligence chatbots can expand access to mental health care by reducing financial barriers. The above indicate that perceived usefulness and perceived ease of use refer to personal advantages of the AI mental health chatbots.

After all, we expected a two-factor model for our scale (i.e., technical and personal advantages).

Figure 1 illustrates the process undertaken to develop and validate the Artificial Intelligence in Mental Health Scale (AIMHS). We followed the suggested guidelines [48,49,50] to develop and validate the AIMHS. Initially, a comprehensive literature review was conducted to identify existing instruments, scales and tools that assess attitudes towards AI (e.g., [20,21,22,24,26,28,29,45,46,51,52]). The relevance and applicability of items from these instruments were critically evaluated to inform the development of the AIMHS. Moreover, we examined instruments, scales and tools that measure mental health issues (e.g., [53,54,55,56,57,58,59,60]). After all, an initial pool of 24 items was generated to capture attitudes towards the use of AI chatbots for mental health support. The initial item pool was structured around the two factors (i.e., technical and personal advantages) that we explain in detail above.

Subsequently, content validity was assessed through consultation with a panel of ten experts from diverse professional backgrounds, including AI experts, psychiatrists, psychologists, therapists, and digital health experts. The panel comprised professionals from diverse domains to ensure that the content validity of the items was evaluated from multiple perspectives, encompassing technical, clinical, and behavioral dimensions. This diversity allowed for a holistic appraisal of the AIMHS, ensuring that it captured the intended constructs across technical, clinical, and behavioral domains. We used the expert panel to assess the content validity of the initial set of 24 items. Each item was rated as “not essential,” “useful but not essential,” or “essential.” The Content Validity Ratio (CVR) was calculated using the following formula, where N represents the total number of experts and n denotes the number who rated an item as “essential.”$$Content\;validity\;ratio=\frac{n-\frac{N}{2}}{\frac{N}{2}}$$

Following suggested guidelines [61], items with a CVR below 0.80 were excluded, resulting in the retention of 14 items.

To evaluate face validity, cognitive interviews [62] were conducted with 15 participants, all of whom demonstrated clear comprehension of the items. Further validation was performed using the item-level face validity index, based on ratings from 15 participants using a four-point Likert scale (1 = not clear to 4 = highly clear). In accordance with suggested guidelines [63], items with an item-level face validity index above 0.80 were retained. All 14 items met this criterion, with item-level face validity index ranging from 0.933 to 1.000.

After the scale’s initial development phase, it comprised 14 items designed to assess attitudes towards the use of AI chatbots for mental health support. Responses were recorded on a five-point Likert scale as follows: 1 (strongly disagree), 2 (disagree), 3 (neither disagree nor agree), 4 (agree), 5 (strongly agree). Appendix A shows the 14 items that were produced after the assessment of the content and face validity of the AIMHS. To reduce acquiescence bias, we employed positive and negative items towards AI chatbots use. In particular, we used seven positive items and seven negative items. Examples of positive items were the following: (1) Artificial intelligence chatbots can demonstrate sufficient social and cultural sensitivity to support diverse populations, (2) Artificial intelligence chatbots can demonstrate better problem-solving skills compared to a human therapist, and (3) Artificial intelligence chatbots can expand access to mental health care by reducing geographic barriers. Examples of negative items were the following: (1) Artificial intelligence chatbots cannot understand people’s emotions, (2) Artificial intelligence chatbots cannot provide correct information on mental health issues, and (3) Artificial intelligence chatbots cannot achieve empathy levels comparable to those of a human therapist.

### 2.2. Participants and Procedure

The study sample consisted of adults aged 18 years or older. Inclusion criteria required participants to engage with AI chatbots, social media platforms, or websites for a minimum of 30 min per day, thereby ensuring a basic level of interaction with digital technologies. Data collection was conducted through an anonymous online survey, developed via Google Forms and disseminated across multiple digital platforms, including Facebook, Instagram, and LinkedIn. To enhance recruitment, a promotional video was additionally produced and shared on TikTok. Data collection took place in October 2025.

In total, 428 individuals completed the questionnaire. Of these, 77.6% (n = 332) were female and 22.4% (n = 96) were male. Higher level of participation among females was expected since females are more willing to complete surveys than males. For instance, a meta-analysis showed that the female-only studies had a higher mean response rate (49.3%) than the male-only studies (28.8%). The impact of gender on the online survey response rate is even higher with female participants showing much higher response rates than male participants [64]. The mean age of participants was 41.1 years (standard deviation; 14.1), with a median of 45 years (range: 18–65). The reported mean daily use of AI chatbots, social media platforms, and websites was 3.5 h (SD = 2.6), with a median of 3 h (range: 30 min–14 h).

### 2.3. Item Analysis

After the scale’s initial development phase, an item analysis was carried out on the 14 generated items. This analysis, based on the full sample (n = 428), examined inter-item correlations, corrected item-total correlations, floor and ceiling effects, skewness, kurtosis, and Cronbach’s alpha (calculated when each individual item was removed) for the 14 items [65]. Inter-item correlations were compared against the recommended range of 0.15 to 0.75 [66], while corrected item-total correlations needed to exceed 0.30 to confirm sufficient discriminatory power [67].

Each item was rated on a five-point Likert scale with anchors: strongly disagree, disagree, neither disagree nor agree, agree, strongly agree. Floor and ceiling effects were identified if over 85% of respondents chose either the lowest (“strongly disagree”) or highest (“strongly agree”) option [68]. Distribution normality was evaluated using skewness (acceptable between −2 and +2) and kurtosis (acceptable between −3 and +3) [69].

An introductory note was provided to participants to describe and clarify the role of AI chatbots in mental health support. Appendix A presents this introductory note. We asked participants to assess their attitudes towards AI chatbots for mental health support by answering the AIMHS.

### 2.4. Construct Validity

To evaluate the construct validity of the AIMHS, we applied both exploratory factor analysis (EFA) and confirmatory factor analysis (CFA). According to the literature, EFA requires a minimum of either 50 participants or five participants per item [70], while CFA typically requires at least 200 participants [65]. Our sample size met these criteria. For stronger psychometric validation, participants were randomly split into two groups: one group of 214 participants for the EFA and another 214 for the CFA. Using independent samples for each factor analysis (i.e., EFA and CFA) strengthened the reliability and validity of our results, with both subsamples surpassing the recommended thresholds.

We began with an EFA to identify the underlying factor structure of the AIMHS, followed by a CFA to test the factor model suggested by the EFA. This stage included the 14 items that remained after the scale’s initial development and item analysis. Before conducting the EFA, data adequacy was confirmed using the Kaiser-Meyer-Olkin (KMO) index and Bartlett’s test of sphericity. Standard acceptability thresholds were applied: KMO values above 0.70 and a statistically significant Bartlett’s test (*p* < 0.05) [69,71]. For the EFA, we used oblique rotation (direct oblimin method in SPSS) since this method allow the factors to correlate as we expected some correlation in our study [65]. Indeed, there was a statistically significant correlation (r = 0.5, *p*-value < 0.001) between the two factors that emerged through validity and reliability analysis. Since the data followed the normal distribution, the maximum likelihood estimator was used [65]. We applied the following acceptability criteria: (a) eigenvalues greater than 1, (b) factor loadings above 0.60, (c) communalities exceeding 0.40, and (d) total explained variance over 65% [69]. Hair et al. argued that all factor loadings should be at least 0.5 and, ideally, at least 0.7 [69]. Reliability was further tested using Cronbach’s alpha, with coefficients above 0.70 considered satisfactory [72].

Subsequently, a CFA was conducted to confirm the AIMHS factor structure. Model fit was assessed through several indices: absolute fit measures (Root Mean Square Error of Approximation [RMSEA] and Goodness of Fit Index [GFI]), relative fit measures (Normed Fit Index [NFI] and Comparative Fit Index [CFI]), and one parsimonious fit measure (ratio of chi-square to degrees of freedom [x^2^/df]). Acceptable thresholds were RMSEA < 0.10, GFI > 0.90, NFI > 0.90, CFI > 0.90, and x^2^/df < 5 [73,74]. Additionally, standardized regression weights between items and factors were calculated.

Moreover, we assessed measurement invariance across demographic subgroups, specifically examining differences by gender, age, and daily use of AI chatbots, social media platforms, and websites. Participants were divided into two groups for age and usage time based on median values. Configural invariance was tested first, followed by metric invariance and then by scalar invariance. We evaluated RMSEA, CFI, and Standardized Root Mean Square Residual (SRMR). For configural invariance, acceptable thresholds defined as RMSEA < 0.10 and CFI > 0.90 [75]. Literature suggest that, for metric invariance, a change of CFI (ΔCFI) from the configural invariance model should be <0.01, a change of RMSEA (ΔRMSEA) should be <0.015, and a change of SRMR (ΔSRMR) should be <0.30. Moreover, for scalar invariance, a change of CFI (ΔCFI) from the metric invariance model should be <0.01, a change of RMSEA (ΔRMSEA) should be <0.015, and a change of SRMR (ΔSRMR) should be <0.10 [76]. For metric invariance, a *p*-value greater than 0.05 indicated invariance [75].

### 2.5. Concurrent Validity

Following the construct validity assessment, a two-factor model with 5 items was established for the AIMHS. Each item was rated on a five-point Likert scale ranging from “strongly disagree” to “strongly agree”. Scores were assigned so that higher values reflected more positive attitudes towards AI chatbots. Thus, scores for positive items were the following: 1 for “strongly disagree”, 2 for “disagree”, 3 for “neither disagree nor agree”, 4 for “agree”, and 5 for “strongly agree”. On the other hand, scores for the negative items were the following: 5 for “strongly disagree”, 4 for “disagree”, 3 for “neither disagree nor agree”, 2 for “agree”, and 1 for “strongly agree”. To obtain the total AIMHS score, we sum the answers in all items and divide the aggregate by total number of answers. Thus, the total AIMHS score ranges from 1 to 5. Similarly, we calculate total score for each factor, and, thus, this score also ranges from 1 to 5. Higher scores indicate more positive attitudes towards AI mental health chatbots.

There are no other scales that measure attitudes towards the use of AI chatbots for mental health support. Thus, we used three other scales to measure concurrent validity of the AIMHS. In particular, we used the Artificial Intelligence Attitude Scale (AIAS-4) [24], the Attitudes Towards Artificial Intelligence Scale (ATAI) [21], and the Short Trust in Automation Scale (S-TIAS) [29].

We used the AIAS-4 [24] to measure general attitude toward AI. The AIAS-4 includes four items such as “I believe that artificial intelligence will improve my life” and “I think artificial intelligence technology is positive for humanity”. Answers are on a 10-point Likert scale from 1 (not at all) to 10 (completely agree). Total score is the average of all the item scores and ranges from 1 to 10. Higher scores indicate more positive attitudes towards AI. We used the valid Greek version of the AIAS-4 [77]. In our study, Cronbach’s alpha for the AIAS-4 was 0.917. We expected a positive correlation between the AIAS-4 and the AIMHS.

We used the ATAI [21] to measure acceptance and fear of AI. The ATAI includes five items; two items measure acceptance of AI, and three items measure fear of AI. The two items for the acceptance of AI are the following: “I trust artificial intelligence” and “Artificial intelligence will benefit humankind”. The three items for the fear of AI are the following: “I fear artificial intelligence”, “Artificial intelligence will destroy humankind” and “Artificial intelligence will cause many job losses”. Answers are on an 11-point Likert scale from 0 (strongly disagree) to 10 (strongly agree). Total score is the average of all the item scores and ranges from 0 to 10. Higher scores in the acceptance subscale indicate higher levels of acceptance of AI, and, thus, more positive attitudes towards AI. Higher scores in the fear subscale indicate higher levels of fear of AI, and, thus, more negative attitudes towards AI. We used the valid Greek version of the ATAI [78]. In our study, Cronbach’s alpha for the acceptance subscale was 0.738, and for the fear subscale was 0.721. We expected a positive correlation between the acceptance subscale and the AIMHS. Also, we expected a negative correlation between the fear subscale and the AIMHS.

We used the S-TIAS [29] to measure trust in AI. The S-TIAS includes three items; “I am confident in the artificial intelligence assistant”, “The artificial intelligence assistant is reliable”, and “I can trust the AI assistant”. Answers are on a 7-point Likert scale from 1 (not at all) to 7 (extremely). Total score is the average of all the item scores and ranges from 1 to 7. Higher scores indicate higher levels of trust in AI, and, thus, more positive attitudes towards AI. We used the valid Greek version of the S-TIAS [79]. In our study, Cronbach’s alpha for the S-TIAS was 0.912. We expected a positive correlation between the S-TIAS and the AIMHS.

### 2.6. Reliability

To assess the internal consistency of the AIMHS, we computed Cronbach’s alpha using the full sample (n = 428). Values above 0.70 for Cronbach’s alpha were regarded as acceptable [72]. We also examined corrected item-total correlations and Cronbach’s alpha if an item was deleted for the final five-item version of the AIMHS, with correlations above 0.30 considered satisfactory [67].

In addition, a test-retest reliability study was carried out with 50 participants who completed the AIMHS twice within a five-day interval. For the five items, Cohen’s kappa was calculated due to the five-point ordinal response format, while a two-way mixed intraclass correlation coefficient (absolute agreement) was used to evaluate the total AIMHS score.

### 2.7. Ethical Issues

The study utilized an anonymous and voluntary data collection process. Before taking part, participants were fully informed about the research aims and procedures and subsequently provided informed consent. Ethical approval was granted by the Institutional Review Board of the Faculty of Nursing, National and Kapodistrian University of Athens (Protocol Approval #01, 14 September 2025). All procedures adhered to the ethical standards of the Declaration of Helsinki [80].

### 2.8. Statistical Analysis

Categorical variables are described using absolute frequencies (n) and percentages, while continuous variables are summarized with measures of central tendency (mean, median) and dispersion (standard deviation, range), along with minimum and maximum values. Normality of scale distributions was evaluated using both statistical tests (Kolmogorov-Smirnov) and graphical methods (Q-Q plots). As all scales showed normal distribution, correlations between them were assessed with Pearson’s correlation coefficients. *p*-values less than 0.05 were considered as statistically significant. We performed CFA with AMOS version 21 (Amos Development Corporation, 2018, Armonk, NY, USA: IBM Corp.). All other analyses were conducted with IBM SPSS 28.0 (IBM Corp. Released 2012. IBM SPSS Statistics for Windows, Version 28.0. Armonk, NY, USA: IBM Corp.).

## 3. Results

### 3.1. Item Analysis

Item analysis included the 14 items developed during the initial phase of the AIMHS. All items demonstrated acceptable psychometric properties, including corrected item–total correlations, inter-item correlations, floor and ceiling effects, skewness, and kurtosis (Table 1). The internal consistency of the scale was high, with a Cronbach’s alpha of 0.888, which decreased upon removal of any single item. Appendix A provides the inter-item correlation matrix for the 14 items.

### 3.2. Exploratory Factor Analysis

The Kaiser-Meyer-Olkin index was 0.861, and Bartlett’s test of sphericity was significant (*p* < 0.001), confirming sample adequacy for EFA. An oblique rotation (direct oblimin in SPSS) was applied to the 14 items (Table 1). Results of the initial EFA are presented in Table 2. Seven items (#1, #2, #5, #10, #11, #13, and #14) exhibited inadequate factor loadings (<0.60) and/or communalities (< 0.40). This analysis yielded three factors explaining 59.758% of the variance. A second EFA was conducted after removing these items. The KMO index was 0.766, and Bartlett’s test remained significant (*p* < 0.001). Table 3 summarizes these results, where two items (#3 and #12) showed insufficient loadings or/and communalities. The second EFA identified two factors accounting for 66.199% of the variance. A third EFA was then performed after excluding items #3 and #12. The KMO index was 0.708, and Bartlett’s test was significant (*p* < 0.001). In this third model, all remaining items (#4, #6, #7, #8, and #9) had factor loadings above 0.649 and communalities above 0.415. The EFA identified two factors explaining 81.280% of the variance, making the third model (Table 4) the optimal structure derived from EFA.

After all, the EFA identified a two-factor five-item model for the AIMHS. These factors were named “technical advantages” and “personal advantages” after consideration of the items that included. In particular, the factor “technical advantages” included the items “Artificial intelligence chatbots cannot achieve empathy levels comparable to those of a human therapist” and “Artificial intelligence chatbots can demonstrate better problem-solving skills compared to a human therapist”. Also, the factor “personal advantages” included the items “Artificial intelligence chatbots can expand access to mental health care by reducing geographic barriers”, “Artificial intelligence chatbots can expand access to mental health care by providing continuous access (24/7 availability)”, and “Artificial intelligence chatbots can expand access to mental health care by reducing financial barriers”.

Cronbach’s alpha for the AIMHS (third model, Table 4) was 0.798, while for the factor “technical advantages” was 0.728, and for the factor “personal advantages” was 0.881.

### 3.3. Confirmatory Factor Analysis

Confirmatory factor analysis was conducted to validate the factor structure of the AIMHS identified through EFA (Figure 2). The goodness-of-fit indices indicated that the two-factor model comprising five items demonstrated a very good fit to the data since x^2^/df was 1.485, RMSEA was 0.048 (90% confidence interval = 0.0001 to 0.122), GFI was 0.989, NFI was 0.987 and CFI was 0.996. Additionally, standardized regression weights between items and factors ranged from 0.63 to 0.92 (*p* < 0.001 in all cases). We found a statistically significant correlation between the two factors (0.5, *p*-value < 0.001).

In conclusion, EFA and CFA identified a two-factor five-item model for the AIMHS. As we described above in detail, the two factors were named “technical advantages” and “personal advantages”.

### 3.4. Measurement Invariance

Table 5 presents the results of the measurement invariance analysis for the two-factor five-item CFA model across gender, age, and daily use of AI chatbots, social media platforms, and websites. The configural, metric and scalar invariance models of AIMS fitted the data well since they fulfilled the critical values. In particular, ΔRMSEA and ΔCFI in all cases were 0.000. Moreover, ΔSRMR values ranged between 0.006 and 0.027 and were acceptable since acceptable values for the metric invariance were less than 0.30, and for the scalar invariance were less than 0.10. Furthermore, metric invariance was supported, as the *p*-values for gender (0.597), age (0.204), and daily use of AI chatbots, social media platforms, and websites (0.261) exceeded the 0.05 threshold. Thus, the two-factor CFA model demonstrated very good fit for all comparisons, including females versus males, younger versus older participants, and those with lower versus higher daily use of AI chatbots, social media platforms, and websites.

### 3.5. Concurrent Validity

We found a positive correlation between the AIAS-4 and the AIMHS (r = 0.405, *p*-value < 0.001), suggesting that participants with more positive attitudes towards AI have more positive attitudes towards AI mental health chatbots.

Moreover, we found a positive correlation between the acceptance subscale of the ATAI and the AIMHS (r = 0.401, *p*-value < 0.001), suggesting that participants with higher levels of acceptance of AI have more positive attitudes towards AI mental health chatbots. On the other hand, we found a negative correlation between the fear subscale of the ATAI and the AIMHS (r = −0.151, *p*-value = 0.002), suggesting that participants with higher levels of fear of AI have more negative attitudes towards AI mental health chatbots.

Additionally, we found a positive correlation between the S-TIAS and the AIMHS (r = 0.450, *p*-value < 0.001), suggesting that participants with higher levels of trust in AI have more positive attitudes towards AI mental health chatbots.

Therefore, the concurrent validity of the AIMHS was satisfactory. Table 6 shows the correlations between the AIMHS, and the AIAS-4, the ATAI and the S-TIAS.

### 3.6. Reliability

Cronbach’s alpha for the AIMHS was 0.798, for the factor “technical advantages” was 0.728, and for the factor “personal advantages” was 0.881.

Corrected item-total correlations for the five items ranged between 0.451 and 0.734, while removal of each single item did not increase Cronbach’s alpha for the AIMHS (Appendix A).

Cohen’s kappa for the five items ranged from 0.760 to 0.848 (*p* < 0.001 in all cases), (Appendix A).

Intraclass correlation coefficient for the AIMHS was 0.938 (95% confidence interval; 0.866 to 0.968, *p* < 0.001).

After all, the reliability of the AIMHS was adequate.

## 4. Discussion

According to the World Health Organization (WHO), the global prevalence of mental disorders continues to rise, posing a significant public health challenge. Recent estimates indicate that approximately one in five individuals worldwide—equivalent to nearly 1.7 billion people—are currently living with a mental disorder, with depressive and anxiety disorders being the most common [81]. Depression affects about 5% of adults globally and remains the leading cause of disability, while anxiety disorders impact roughly 4% of adults [82]. The burden of these conditions is particularly pronounced in low- and middle-income countries, where access to mental health care remains limited and comorbidity between depression and anxiety disorders is common [81].

Usage of mental health chatbots and virtual therapists has increased by 320% between 2020 and 2022. By 2023, around 84% of mental health professionals had either adopted or considered integrating AI tools into their work. By 2032, it is expected that nearly all professionals—approximately 99%—will use AI technologies as part of their routine practice [83].

This surge reflects a broader trend toward digital mental health solutions, driven by accessibility, affordability, and the demand for immediate support. Market projections indicate continued expansion, with the mental health chatbot sector expected to grow from $380 million in 2025 to $1.65 billion by 2033, representing a compound annual growth rate of approximately 20% [84].

To the best of our knowledge, no existing scale measures individuals’ attitudes toward the use of AI chatbots for mental health support. In particular, several scales until now have measured attitudes toward artificial intelligence in general [20,22,24,25]. For instance, the Attitudes Towards Artificial Intelligence Scale measures attitudes towards AI as a single construct, independent of specific contexts or applications [20], while Schepman and Rodway created the General Attitudes towards Artificial Intelligence Scale to measure general attitudes toward AI, separating attitudes toward the positive and negative aspects of AI [22]. In a similar way, the Artificial Intelligence Attitudes Inventory includes two 8-item subscales to measure positive and negative attitudes toward AI [25]. Additionally, the Artificial Intelligence Attitudes Scale is a very brief tool including four items that measure users’ general attitudes toward AI [24]. In addition, several other scales have been developed to assess attitudes toward specific domains of AI [21,26,27,28,29,30]. In particular, Sindermann et al. created the Attitude Towards Artificial Intelligence to measure two specific aspects of attitudes toward AI, i.e., acceptance and fear of AI [21]. Also, Park et al. considered that understanding employees’ attitudes toward the application of AI is essential for its successful integration within an organization, and, thus, created a scale to measure attitudes towards AI at work [26]. Kieslich et al. considered the growing concern about the impact of AI applications on individuals and society, and, therefore, developed and validated a scale to measure attitudes toward the threats of AI over three distinct AI domains (medical treatment, job recruitment, and loan origination) [27]. Moreover, McGrath et al. validated the Trust in Automation Scale across a range of AI applications to measure human trust in AI [29], while Wang and Wang developed and validated a scale to capture general public anxiety toward AI development [28]. Finally, the Attitude Scale Towards the Use of Artificial Intelligence Technologies in Nursing was specifically developed to measure nurses’ attitudes toward the AI technology in clinical settings [30].

Therefore, we developed and validated the Artificial Intelligence in Mental Health Scale to measure individuals’ attitudes toward the use of AI-based chatbots for mental health support and to address this gap in the field. Our comprehensive assessment of validity and reliability indicates that the AIMHS demonstrates adequate psychometric soundness as a measure of attitudes toward the use of AI-based chatbots for mental health support. Comprising five items and requiring only a few minutes to administer, the AIMHS offers a concise and user-friendly option while maintaining robust measurement properties.

We adhered to the recommended guidelines [48,49,50] for the development and validation of the AIMHS. Following an extensive literature review on instruments, scales and tools that assess attitudes towards AI, and mental health issues, we initially generated 24 items intended to measure attitudes towards AI mental health chatbots. The content validity of these items was evaluated using the content validity ratio, resulting in the removal of 10 items and leaving 14 items for further analysis. Face validity was assessed through cognitive interviews and the calculation of the item-level face validity index, after which all 14 items advanced to the next stage of evaluation. Subsequently, item analysis was conducted, leading to adequate indices for the 14 items. Consequently, 14 items were retained for construct validity testing. Exploratory factor analysis revealed that nine items exhibited unacceptable factor loadings and communalities, prompting their removal. The remaining five items demonstrated satisfactory values in the EFA, which identified two factors. As we expected, factor analysis revealed a two-factor model for AIMHS; technical advantages and personal advantages. Confirmatory factor analysis further supported the two-factor five-item structure identified by the EFA. To further assess validity, three validated scales were employed to examine concurrent. In particular, we used the AIAS-4, the ATAI, and the S-TIAS. Statistically significant correlations were observed between the AIMHS and these scales, indicating adequate construct validity.

### Limitations

Our study had several limitations. First, this study was conducted within a single country using a convenience sampling method. For instance, the proportion of male participants was notably lower than that of the general population resulting on a heavy gender imbalance in this study. Moreover, we employed a sample of daily technology users that could introduce significant selection bias in this study. Consequently, the results and normative scores may not be widely generalized to the broader population. Future research should aim to include more representative and diverse samples to further validate the AIMHS. Special attention should be given to technology-hesitant individuals who are a critical target group for accessible mental health solutions such as the AI mental health chatbots. Despite these limitations, the psychometric evaluation remains strong, as the sample size met all required standards. Moreover, our findings supported configural measurement invariance and metric invariance considering several demographic variables. However, it is important for future studies to explore the psychometric properties of the AIMHS across various populations and cultural contexts. Second, we employed a cross-sectional design to evaluate the validity of the AIMHS. However, since individuals’ attitudes may shift over time, longitudinal studies are needed to better understand individuals’ attitudes toward the use of AI chatbots for mental health support. Third, while we conducted a comprehensive psychometric analysis of the AIMHS, future researchers may consider additional methods—such as assessing divergent, criterion, and known-groups validity—to further examine its reliability and validity. Fourth, as this was the initial validation of the AIMHS, we did not attempt to establish a cut-off score. Future investigations could explore threshold values to help differentiate between participant groups. Fifth, self-report instruments are inherently susceptible to social desirability bias in studies like this one. However, since our study was conducted anonymously and participation was entirely voluntary, we believe that the likelihood of such bias influencing responses was minimal. Sixth, we used three scales to examine the concurrent validity of the AIMHS: (1) Artificial Intelligence Attitude Scale, (2) Attitudes Towards Artificial Intelligence Scale, and (3) Short Trust in Automation Scale. The concurrent validity of the AIMHS can be further explored by using several other scales such as the Attitudes Towards Artificial Intelligence Scale [20], the General Attitudes Towards Artificial Intelligence Scale (GAAIS) [22], and the Artificial Intelligence Attitudes Inventory (AIAI) [25]. Sixth, the technical advantages factor may exhibit instability due to its composition of only two items. The presence of factors with a limited number of items does not inherently compromise the psychometric robustness of a scale, but this issue should be addressed in future studies. Our reliability and validity analyses showed that the two factors of the AIMS have satisfactory psychometric properties. However, future research should assess the reliability and validity of the AIMHS, with particular emphasis on the technical advantages factor. Seventh, the AIMHS ultimately captures two key dimensions of attitudes toward AI mental health chatbots: technical advantages and personal advantages. Given that AIMHS represents the first validated instrument specifically designed to measure these attitudes, it is reasonable to assume that additional domains—such as emotion recognition, stigma reduction, and emergency response capabilities—may also influence individuals’ perceptions of such technologies. Consequently, future research should rigorously examine the psychometric properties of AIMHS across diverse populations and contexts to ensure its reliability and validity. Furthermore, the development and validation of complementary scales targeting these emerging domains would be of considerable importance for advancing our understanding of user attitudes toward AI mental health chatbots and informing their ethical and effective integration into mental health care. Another limitation of this study is the fact that the AIMHS includes one negatively worded item (i.e., Artificial intelligence chatbots cannot achieve empathy levels comparable to those of a human therapist), while the other four items are positively worded. Thus, the scale probably creates a positive framing bias and may lack sensitivity to detect ambivalence or negative attitudes. Future similar scales may consider incorporating additional negatively worded items. Additionally, we performed the test-retest study over a very short day interval (five days), and, thus, it would be difficult to account for the potential variability of individuals’ attitudes toward AI mental health chatbots. Since media exposure and public disclosure may affect these attitudes it would be of great importance to examine the temporal stability of the AIMHS over longer periods. We should recognize that we did not examine the predictive validity of the AIMHS, and, thus, we cannot evaluate the ability of the scale to predict key behavioral outcomes into healthcare such as actual usage intention or engagement. Since prediction validity is an essential form of validity future studies should consider to examine it by providing more valid results regarding the psychometric properties of the AIMHS. Moreover, we should emphasize the fact that the 5-item AIMHS although it is a user-friendly and easy to use scale may be too superficial to inform the design and execution of user-centered interventions regarding AI mental health chatbots. Thus, further studies should be conducted to examine the psychometric properties of the AIMHS especially in situations where scholars evaluate the effectiveness of mental health interventions. We should pay special attention to the interpretation of the findings of the concurrent validity of the AIMHS. The moderate correlations between the AIMHS and the general artificial intelligence attitude scales denote that the AIMHS could measure a related but distinct construct such as attitudes toward AI mental health chatbots. However, the weak correlation between the AIMHS and the fear of artificial intelligence subscale raises concerns that the AIMHS may be incapable of capturing ethical concerns and salient negative attitudes towards AI mental health chatbots. Finally, the AIMHS as well as the other scales we used in our study were self-report instruments, and, thus, an information bias is probable due to some participant subjectivity.

## 5. Conclusions

Given the rapid integration of AI into mental health care, assessing attitudes toward AI-based chatbots is essential for understanding their acceptance and potential impact. The AIMHS is a brief tool focused on perceived advantages, and that other important domains (e.g., ethical concerns, stigma, emergency response) will need complementary measures. User perceptions regarding trust, privacy, perceived usefulness, and emotional safety significantly influence adoption and sustained engagement. If individuals perceive chatbots as helpful, trustworthy, and easy to use, they are more likely to integrate them into their mental health routines.

However, currently, there is no widely validated scale specifically for measurement of attitudes toward AI mental health chatbots. To address this gap, we developed and validated the Artificial Intelligence in Mental Health Scale. Measuring attitudes toward these technologies can inform the design of user-centered interventions, guide policy development, and ensure that digital mental health solutions are implemented in a manner that is both effective and ethically responsible. Even the most advanced chatbot will fail to deliver benefits if users are skeptical or resistant. Measuring attitudes helps identify barriers such as privacy concerns, lack of empathy perception, or fear of misdiagnosis. Understanding public attitudes informs regulatory frameworks, ethical guidelines, and implementation strategies for digital mental health interventions. Developers can use attitude data to improve user experience, cultural sensitivity, and trust-building features in AI systems.

After thoroughly evaluating its reliability and validity, we determined that the AIMHS is a concise and accessible instrument with satisfactory psychometric qualities. The results indicate that AIMHS functions as a two-factor scale comprising five items, designed to gauge individuals’ perceptions of AI-based mental health chatbots. Considering the study’s limitations, we suggest translating and validating the AIMHS across various languages and demographic groups to further examine its consistency and accuracy. Overall, the AIMHS shows potential as a valuable resource for assessing attitudes toward AI mental health chatbots and could assist policymakers, educators, healthcare providers, and other stakeholders in enhancing mental health services. For example, both researchers and practitioners can use the AIMHS to evaluate the effectiveness of mental health interventions by measuring attitudes before and after chatbot interaction to assess changes. Furthermore, researchers in all disciplines may now use a valid tool such as AIMHS to explore associations between attitudes and variables like usage intention, mental health outcomes, or demographics. For instance, the AIMHS may be used to predict intention to use AI chatbots for mental health support, examine factors influencing acceptance, and evaluate interventions aimed at improving acceptance.

Assessing individuals’ acceptance of AI-based mental health chatbots is critical for understanding their potential integration into mental healthcare services. Acceptance influences both initial adoption and sustained engagement, which are essential for achieving therapeutic benefits. Measuring acceptance through validated scales enables researchers to identify key determinants such as perceived usefulness, ease of use, trust, and privacy concerns. These insights can inform the design of user-centered interventions, guide ethical implementation, and support policy development aimed at promoting equitable access to digital mental health solutions.

## Figures and Tables

**Figure 1 healthcare-13-03269-f001:**
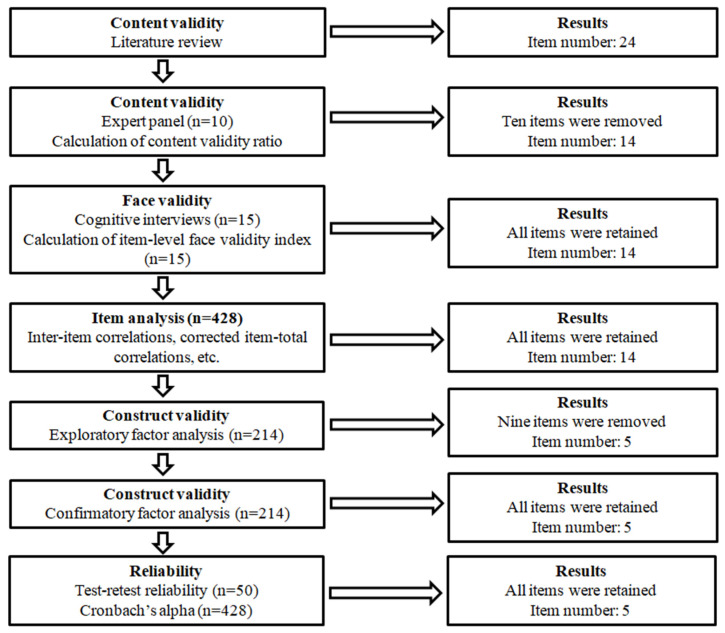
Development and validation of the Artificial Intelligence in Mental Health Scale.

**Figure 2 healthcare-13-03269-f002:**
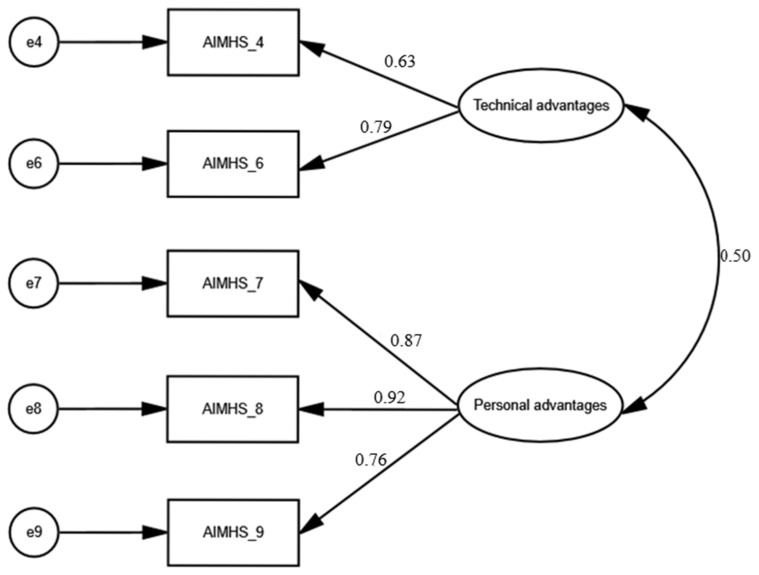
Confirmatory factor analysis of the Artificial Intelligence in Mental Health Scale.

**Table 1 healthcare-13-03269-t001:** Descriptive statistics, corrected item-total correlations, floor and ceiling effects, skewness, kurtosis, and Cronbach’s alpha (when a single item was deleted) for the 14 items that were produced after the initial development phase of the Artificial Intelligence in Mental Health Scale (n = 428).

Artificial Intelligence Chatbots …	Mean (Standard Deviation)	Corrected Item-Total Correlation	Floor Effect (%)	Ceiling Effect (%)	Skewness	Kurtosis	Cronbach’s Alpha if Item Deleted	Item Exclusion or Retention
1. cannot understand people’s emotions	2.36 (0.94)	0.543	19.4	0.5	0.21	−0.67	0.882	Retained
2. can demonstrate sufficient social and cultural sensitivity to support diverse populations	2.91 (0.94)	0.587	7.7	1.6	−0.28	−0.52	0.879	Retained
3. cannot provide correct information on mental health issues	2.77 (0.79)	0.566	7.0	0.5	−0.37	0.13	0.881	Retained
4. cannot achieve empathy levels comparable to those of a human therapist	1.76 (0.79)	0.491	42.5	0.1	0.85	0.41	0.884	Retained
5. cannot appropriately respond to mental health emergencies (e.g., suicidal thoughts)	2.30 (1.02)	0.492	25.9	0.5	0.26	−0.93	0.884	Retained
6. can demonstrate better problem-solving skills compared to a human therapist	1.95 (0.81)	0.541	32.2	0.5	0.49	−0.11	0.882	Retained
7. can expand access to mental health care by reducing geographic barriers	3.29 (0.96)	0.672	5.8	4.7	−0.70	0.01	0.875	Retained
8. can expand access to mental health care by providing continuous access (24/7 availability)	3.43 (0.90)	0.708	4.2	6.5	−0.75	0.52	0.874	Retained
9. can expand access to mental health care by reducing financial barriers	3.44 (0.93)	0.611	4.4	7.9	−0.70	0.40	0.878	Retained
10. can eliminate social stigma concerns since users maintain their privacy	3.08 (1.02)	0.460	7.5	4.2	−0.33	−0.69	0.886	Retained
11. cannot improve people’s mood	3.03 (0.80)	0.501	4.0	0.9	−0.41	0.12	0.883	Retained
12. can offer up to date information on mental health issues	3.07 (0.82)	0.559	4.7	0.9	−0.54	0.13	0.881	Retained
13. can empower people by providing them with personalized support	2.72 (0.92)	0.625	9.6	0.9	−0.09	−0.60	0.878	Retained
14. cannot enhance early detection for mental health conditions	2.69 (0.90)	0.582	9.8	0.7	−0.10	−0.57	0.880	Retained

**Table 2 healthcare-13-03269-t002:** Exploratory factor analysis (first model) using oblique rotation (direct oblimin method in SPSS) for the Artificial Intelligence in Mental Health Scale (n = 214).

Artificial Intelligence Chatbots …	Factors			Communalities
	First (Items #4, #6)	Second (Items #7, #8, #9)	Third (Items #3, #12)	
1. cannot understand people’s emotions	0.344	0.578	0.399	0.463
2. can demonstrate sufficient social and cultural sensitivity to support diverse populations	0.536	0.437	0.491	0.467
3. cannot provide correct information on mental health issues	0.361	0.471	0.601	0.403
4. cannot achieve empathy levels comparable to those of a human therapist	0.226	0.831	0.263	0.536
5. cannot appropriately respond to mental health emergencies (e.g., suicidal thoughts)	0.410	0.537	0.348	0.348
6. can demonstrate better problem-solving skills compared to a human therapist	0.302	0.745	0.381	0.488
7. can expand access to mental health care by reducing geographic barriers	0.815	0.425	0.472	0.655
8. can expand access to mental health care by providing continuous access (24/7 availability)	0.933	0.388	0.489	0.730
9. can expand access to mental health care by reducing financial barriers	0.792	0.233	0.486	0.592
10. can eliminate social stigma concerns since users maintain their privacy	0.511	0.277	0.544	0.341
11. cannot improve people’s mood	0.355	0.294	0.483	0.250
12. can offer up to date information on mental health issues	0.403	0.280	0.778	0.404
13. can empower people by providing them with personalized support	0.483	0.586	0.552	0.481
14. cannot enhance early detection for mental health conditions	0.438	0.549	0.552	0.450
Eigenvalues	5.604	1.632	1.130	
% of variance	40.030	11.655	8.073	
Cumulative % of variance	40.030	51.685	59.758	

Values express factors loadings unless otherwise is indicated.

**Table 3 healthcare-13-03269-t003:** Exploratory factor analysis (second model) using oblique rotation (direct oblimin method in SPSS) for the Artificial Intelligence in Mental Health Scale (n = 214).

Artificial Intelligence Chatbots …	Factors		Communalities
	First (Items #7, #8, #9)	Second (Items #4, #6)	
1. cannot provide correct information on mental health issues	0.370	0.473	0.330
2. cannot achieve empathy levels comparable to those of a human therapist	0.248	0.775	0.438
3. can demonstrate better problem-solving skills compared to a human therapist	0.321	0.806	0.445
4. can expand access to mental health care by reducing geographic barriers	0.811	0.385	0.617
5. can expand access to mental health care by providing continuous access (24/7 availability)	0.938	0.368	0.714
6. can expand access to mental health care by reducing financial barriers	0.783	0.208	0.568
7. can offer up to date information on mental health issues	0.391	0.294	0.292
Eigenvalues	3.211	1.423	
% of variance	45.870	20.329	
Cumulative % of variance	45.870	66.199	

Values express factors loadings unless otherwise is indicated.

**Table 4 healthcare-13-03269-t004:** Exploratory factor analysis (third model) using oblique rotation (direct oblimin method in SPSS) for the Artificial Intelligence in Mental Health Scale (n = 214).

Artificial Intelligence Chatbots …	Factors		Communalities
	First (Items #4, #6)	Second (Items #7, #8, #9)	
1. cannot achieve empathy levels comparable to those of a human therapist	0.982	0.215	0.415
2. can demonstrate better problem-solving skills compared to a human therapist	0.649	0.293	0.420
3. can expand access to mental health care by reducing geographic barriers	0.353	0.799	0.606
4. can expand access to mental health care by providing continuous access (24/7 availability)	0.314	0.958	0.710
5. can expand access to mental health care by reducing financial barriers	0.161	0.771	0.556
Eigenvalues	2.696	1.368	
% of variance	53.922	27.357	
Cumulative % of variance	53.922	81.280	

Values express factors loadings unless otherwise is indicated.

**Table 5 healthcare-13-03269-t005:** Measurement invariance for the two-factor five-item model of the Artificial Intelligence in Mental Health Scale with respect to gender, age, and daily use of AI chatbots, social media platforms, and websites.

Variable	Levels of Measurement Invariance	RMSEA	CFI	SRMR	ΔRMSEA	ΔCFI	ΔSRMR
Gender	Configural	<0.001	1.000	0.026			
	Metric	<0.001	1.000	0.035	0.000	0.000	0.009
	Scalar	<0.001	1.000	0.050	0.000	0.000	0.015
Age	Configural	<0.001	1.000	0.016			
	Metric	<0.001	1.000	0.043	0.000	0.000	0.027
	Scalar	<0.001	1.000	0.049	0.000	0.000	0.006
Daily use of artificial intelligence chatbots, social media platforms, and websites	Configural	<0.001	1.000	0.019			
	Metric	<0.001	1.000	0.040	0.000	0.000	0.021
	Scalar	<0.001	1.000	0.061	0.000	0.000	0.021

CFI: Comparative Fit Index; RMSEA: Root Mean Square Error of Approximation; SRMR: Standardized Root Mean Square Residual.

**Table 6 healthcare-13-03269-t006:** Correlations between the Artificial Intelligence in Mental Health Scale, and the Artificial Intelligence Attitude Scale, the Attitudes Towards Artificial Intelligence Scale, and the Short Trust in Automation Scale (n = 428).

	Artificial Intelligence in Mental Health Scale	
	Pearson’s Correlation Coefficient	*p*-Value
Artificial Intelligence Attitude Scale	0.405	<0.001
Attitudes Towards Artificial Intelligence Scale (acceptance subscale)	0.401	<0.001
Attitudes Towards Artificial Intelligence Scale (fear subscale)	−0.151	0.002
Short Trust in Automation Scale	0.450	<0.001

## Data Availability

The original data presented in the study are openly available in FigShare at https://doi.org/10.6084/m9.figshare.30556751.

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
