# Peer review of "Development and Validation of the Artificial Intelligence in Mental Health Scale: Application for AI Mental Health Chatbots"

_healthcare, 2025, doi:10.3390/healthcare13243269_

Round 1

Reviewer 1 Report

Comments and Suggestions for Authors

Thank you very much for inviting me to review the manuscript titled “
Attitudes toward the use of Artificial Intelligence Chatbots for Mental Health Support: Artificial Intelligence in Mental Health Scale.” The manuscript has the potential to contribute to the scientific community.  However, several concerns are observed

  1. Title is informative but broad. Also, please adhere to the STROBE guidelines
  2. Abstract is exhaustive. Rationale needs to be enhanced. Too much description about tools leads to a lack of focus on the main summary. Results also require some important numerical results. Also, the conclusion is non-specific
  3. Introduction: Including the updated references adds value to the manuscript. However, it's too lengthy (not in a typical scholarly style).
  4. Also, the manuscript is missing a strong critical gaps analysis (with existing) and conceptualization frameworks.
  5. Too many definitions are not required for an AI chatbot.
  6. Methods: This section is another major concern. Lack of clear item development. Authors state 24 items → 14 after CVR → 5 after EFA/CFA. But the content of the removed items is unclear (e.g., why those with high item-total correlations were later removed in EFA.
  7. The expert panel description needs elaboration, as it is critical.
  8. The participants' recruitment process needs more clarification, and it leads to selection bias. It is confirmed by the heavily skewed gender of the study.
  9. Generally, loading cutoff >0.60 is too high for new scales. Please provide updated references for this.
  10. Results: What is the reason for removing some items, with good psychometric properties (Table 1)?
  11. There is no need to repeat what is emphasized in the introduction.
  12. Also, the discussion did not reflect the risk of bias due to the methodology of the current study.
  13. No comparison with existing AI attitude scales—critical for establishing a unique contribution of the current study.
  14. This study has several limitations. All must be acknowledged or else it may mislead the readers.

Reviewer 2 Report

Comments and Suggestions for Authors

I see clear value in developing a measure of attitudes toward AI chatbots. However, the manuscript places substantial emphasis on the positive features of chatbots while giving relatively limited attention to their negative aspects. AI chatbots can function as psychological stressors, generate hallucinations, and create confusion or misinformation. Given this consideration, the authors should more explicitly integrate these potential downsides into the conceptual framework. This is particularly important because the current study focuses solely on validating the measure without examining its correlations with indicators of functioning or well-being that might reflect these negative dimensions.

Additionally, the assessment of measurement invariance appears incomplete. The manuscript reports only configurial invariance results. To claim that the scale operates equivalently across groups, the authors must also test and report metric and scalar invariance at minimum. Although the authors briefly mention that "metric invariance was supported" based on p-values for gender, age, and frequency of AI chatbot use, the provided information does not permit readers to evaluate this claim. Simply reporting p-values exceeding .05 is insufficient; full model-comparison statistics (e.g., ΔCFI, ΔRMSEA) and clearly labeled invariance tests for each level (configurial, metric, and scalar) are needed.

Finally, while the manuscript presents evidence of convergent validity, it does not adequately address discriminant validity. The scale appears conceptually proximate to general attitudes toward AI or technology more broadly, yet the manuscript does not convincingly distinguish this construct from established measures. The authors briefly allude to this issue, but the discussion remains underdeveloped. More theoretical and empirical clarification is needed to justify why this new measure is necessary and to demonstrate how it uniquely contributes beyond existing attitude constructs.

Reviewer 3 Report

Comments and Suggestions for Authors

The authors have taken a valuable first step in a crucial field. The AIMHS is a promising, brief tool for measuring specific positive perceptions of AI chatbots in mental health, particularly regarding accessibility. However, in its current form, it should not be presented as a comprehensive measure of "attitude." Significant conceptual and methodological concerns suggest that it captures a limited, primarily utilitarian, subset of the attitude construct. Addressing these points would substantially strengthen the manuscript and provide a more scientifically rigorous foundation for future research.

• The scale's conceptual foundation is problematic as it defines attitude as a multi-component construct but the final instrument predominantly measures cognitive beliefs about utility and accessibility, aligning more with the Perceived Usefulness dimension of the Technology Acceptance Model than with a holistic attitudinal construct that should encompass affective and behavioral intention elements, thereby risking construct underrepresentation.

• The two-factor solution is conceptually and psychometrically weak; the Technical Advantages factor is unstable, comprising only two items that conflate a perceived limitation (inability to achieve empathy) with a perceived strength (superior problem-solving), and a two-item factor is generally considered psychometrically fragile and unreliable for drawing substantive conclusions.

• The item reduction process while statistically guided but may have stripped the scale of its conceptual richness as the elimination of nine items removed core content related to emotion understanding, emergency response, and stigma potentially leaving a instrument that measures a narrow view of perceived advantages rather than the full spectrum of public attitudes toward AI in mental health.

• The evidence for concurrent validity is mixed and requires reinterpretation; the modest correlations with general AI attitude scales suggest the AIMHS measures a related but distinct construct but the critically weak correlation with the fear of AI subscale raises concerns that the scale fails to capture salient negative attitudes and ethical concerns, which are paramount in mental health contexts.

• The failure to establish scalar measurement invariance is a major methodological omission, as without it, comparing mean scores across demographic groups like gender and age is statistically invalid severely limiting the scale's utility for the group comparisons the authors themselves suggest in their discussion of future applications.

• The sample composition, being a convenience sample of daily technology users with a heavy gender imbalance, introduces significant selection bias, meaning the scale's properties and normative scores may not generalize to the broader population, including technology-hesitant individuals who are a critical target for accessible mental health solutions.

• The language used to describe psychometric properties, such as excellent reliability and strong validity, is an overstatement given a Cronbach's alpha of 0.798 for the total scale and 0.728 for a two-item subscale which are adequate but not robust, and validity evidence that has notable caveats as previously mentioned.

• The scale's focus on advantages creates a positive framing bias and may lack sensitivity to detect ambivalence or negative attitudes, which is a critical flaw for a tool intended to assess barriers to implementation; this is exacerbated by the removal of negatively worded items during the factor analysis.

• The test-retest reliability over a very short five-day interval does not account for the potential volatility of attitudes toward emerging technologies, which can be influenced by media exposure and public discourse, and therefore provides limited evidence for the temporal stability of the construct over a meaningful period.

• The discussion lacks critical reflection on the significant trade-off between brevity and comprehensiveness, failing to acknowledge that the final 5-item scale, while user-friendly but may be too superficial to inform the design of user-centered interventions or ethical guidelines as the authors propose.

• There is a disconnect between the stated aim of measuring attitudes for integration into healthcare and the scale's apparent inability to predict key behavioral outcomes like actual usage intention or engagement; the predictive validity of the scale remains entirely unknown and is a crucial area for future validation.

Round 2

Reviewer 1 Report

Comments and Suggestions for Authors

Thanks for making the revisions. However, please find the following comments.

1. The title must follow the STROBE Guidelines.

2. Please make sure that the abstract is limited to healthcare, MDPI word count, without removing essential details.

Author Response

Reviewer 1

Dear Reviewer,

Thank you again very much for the peer review of the manuscript. Thank you for your comments, which have improved the quality of the manuscript. We have addressed all the comments (highlighted with the "Track Changes" function) in the revised text. Also, we make changes in the manuscript according to the other Reviewers’ instructions.

Please, find below an item-by-item answer to your comments. Hoping the revised manuscript fulfils the journal’s standards, we thank you for your courtesy.

We trust that these revisions demonstrate our commitment to rigor and transparency, and we kindly ask you to take these improvements into consideration.

We are looking forward to your response.

Yours sincerely,

The authors

Thanks for making the revisions. However, please find the following comments.

Comment

  1. The title must follow the STROBE Guidelines.

Answer: done

Dear Reviewer, we follow the STROBE Guidelines by further describe in the title the aim of our study. The revised title is presented below for your review. Kindly confirm whether it meets your approval. We are fully prepared to incorporate any adjustments or recommendations you may provide.

Development and Validation of the Artificial Intelligence in Mental Health Scale: Application for AI Mental Health Chatbots

Comment

  1. Please make sure that the abstract is limited to healthcare, MDPI word count, without removing essential details.

Answer: done

Dear Reviewer, research articles in the journal “Healthcare” should have a structured abstract of around 250 words. Our previous abstract has 395 words. Thus, we remove text. We remove 122 words, and now the revised abstract has 273 words. Please, see the updated abstract.

We consider your previous comments in the first round of revision regarding the Abstract (i.e., Abstract is exhaustive. Rationale needs to be enhanced. Too much description about tools leads to a lack of focus on the main summary. Results also require some important numerical results. Also, the conclusion is non-specific).

Reviewer 3 Report

Comments and Suggestions for Authors

Thank you for your careful and thoughtful revisions. You have substantially improved the manuscript by clarifying that the AIMHS primarily assesses perceived advantages (technical and personal) of AI mental health chatbots, adding appropriate measurement invariance analyses, and expanding the Limitations section to address content coverage, sample characteristics, short test–retest interval, lack of predictive validity, and the implications of a brief 5-item scale. My remaining comments are minor and mainly editorial: I recommend a light language edit for grammar/phrasing and adding one simple, explicit sentence in the Introduction or Conclusion noting that the AIMHS is a brief tool focused on perceived advantages, and that other important domains (e.g., ethical concerns, stigma, emergency response) will need complementary measures.

Author Response

Reviewer 3

Dear Reviewer,

Thank you for your careful and thoughtful revisions. You have substantially improved the manuscript by clarifying that the AIMHS primarily assesses perceived advantages (technical and personal) of AI mental health chatbots, adding appropriate measurement invariance analyses, and expanding the Limitations section to address content coverage, sample characteristics, short test–retest interval, lack of predictive validity, and the implications of a brief 5-item scale. My remaining comments are minor and mainly editorial: I recommend a light language edit for grammar/phrasing and adding one simple, explicit sentence in the Introduction or Conclusion noting that the AIMHS is a brief tool focused on perceived advantages, and that other important domains (e.g., ethical concerns, stigma, emergency response) will need complementary measures.

We add the following sentence in the first paragraph of conclusions.

The AIMHS is a brief tool focused on perceived advantages, and that other important domains (e.g., ethical concerns, stigma, emergency response) will need complementary measures.

We edit for grammar/phrasing the manuscript. Please see the following changes as examples.

INTRODUCTION

  1. 1. We rewrite this sentence:

“As AI technologies continue to proliferate across diverse sectors, their integration into mental health services has emerged as a significant area of inquiry.”

as follows: “As AI technologies continue to expand across diverse sectors, their integration into mental health services has become an increasingly important area of investigation.”

  1. 2. We rewrite this sentence:

“Recent research has increasingly focused on the use of AI chatbots as innovative tools for digital mental health interventions.”

as follows: “Recent research has increasingly examined the use of AI chatbots as innovative tools for delivering digital mental health interventions.”

  1. 3. We rewrite this sentence:

“These chatbots utilize machine learning algorithms and natural language processing to interpret user input and generate contextually appropriate responses, thereby enabling interactions that closely resemble human conversation.”

as follows: “These chatbots employ machine learning algorithms and natural language processing to interpret user input and produce contextually appropriate responses, thereby enabling interactions that more closely resemble human conversation.”

  1. 4. We rewrite this sentence:

“Empirical studies investigating the effectiveness of AI-based interventions have yielded encouraging findings.”

as follows: “Empirical studies examining the effectiveness of AI-based interventions have produced encouraging results.”

  1. 5. We rewrite this sentence:

“AI chatbots may be particularly well-suited for young adults, a demographic that frequently utilizes online messaging platforms with interfaces resembling those of chatbot applications.”

as follows: “AI chatbots may be particularly suitable for young adults, a demographic that frequently uses online messaging platforms with interfaces similar to those of chatbot applications.”

  1. 6. We rewrite this sentence:

“Although preliminary findings support the effectiveness of AI chatbots and highlight their potential utility, empirical research investigating individuals’ attitudes toward these technologies remains limited.”

as follows: “Although preliminary findings support the effectiveness of AI chatbots and highlight their potential utility, empirical research on individuals’ attitudes toward these technologies remains limited.”

  1. 7. We rewrite this sentence:

“Until now, several scales have developed to measure attitudes towards AI as a …”

as follows: “To date, several scales have been developed to measure attitudes toward AI as a …

  1. 8. We rewrite this sentence:

“Additionally, several other scales have produced to assess attitudes towards specific domains of AI…”

as follows: “Additionally, several other scales have been developed to assess attitudes toward specific domains of AI…

  1. 9. We rewrite this sentence:

“However, to the best of our knowledge, there is no scale that measures individuals’ attitudes toward the use of AI-based chatbots for mental health support.”

as follows: “However, to the best of our knowledge, no existing scale measures individuals’ attitudes toward the use of AI-based chatbots for mental health support.”

DISCUSSION

  1. 1. We rewrite this sentence:

“The burden of these conditions is particularly pronounced in low- and middle-income countries, where access to mental health care remains limited and comorbidity between depression and anxiety disorders is frequent.”

as follows: “The burden of these conditions is particularly pronounced in low- and middle-income countries, where access to mental health care remains limited and comorbidity between depression and anxiety disorders is common.”

  1. 2. We rewrite this sentence:

“Looking ahead to 2032, it's expected that nearly all—99%—will be using AI technologies as part of their everyday practice.”

as follows: “By 2032, it is expected that nearly all professionals—approximately 99%—will use AI technologies as part of their routine practice.”

  1. 3. We rewrite this sentence:

“To the best of our knowledge, there is no scale that measures individuals’ attitudes toward the use of AI chatbots for mental health support.”

as follows: “To the best of our knowledge, no existing scale measures individuals’ attitudes toward the use of AI chatbots for mental health support.”

  1. 4. We rewrite this sentence:

“On the other hand, several other scales have developed to assess attitudes towards specific domains of AI.”

as follows: “In addition, several other scales have been developed to assess attitudes toward specific domains of AI.”

  1. 5. We rewrite this sentence:

“Therefore, we developed and validated the Artificial Intelligence in Mental Health Scale to measure individuals’ attitudes toward the use of AI-based chatbots for mental health support to fill the gap in this research field.”

as follows: “Therefore, we developed and validated the Artificial Intelligence in Mental Health Scale to measure individuals’ attitudes toward the use of AI-based chatbots for mental health support and to address this gap in the field.”

  1. 6. We rewrite this sentence:

“Future similar scales may consider including more negatively worded items.”

as follows: “Future similar scales may consider incorporating additional negatively worded items.”